# Intra-Tumoral CD8+:CD3+ Lymphocyte Density Ratio in Appendix Cancer Is a Tumor Volume- and Grade-Independent Predictor of Survival

**DOI:** 10.3390/cancers17030542

**Published:** 2025-02-06

**Authors:** Chelsea Knotts, Hyun Park, Christopher Sherry, Rose Blodgett, Catherine Lewis, Ashten Omstead, Kunhong Xiao, William LaFramboise, David L. Bartlett, Neda Dadgar, Ajay Goel, Ali H. Zaidi, Patrick L. Wagner

**Affiliations:** 1Allegheny Health Network Cancer Institute, Pittsburgh, PA 15212, USA; knotts.chelsea@ahn.org (C.K.); hyunyoung.park@ahn.org (H.P.); christopher.sherry2@ahn.org (C.S.); rose.blodgett@ahn.org (R.B.); catherine.lewis@ahn.org (C.L.); ashten.omstead@ahn.org (A.O.); kevin.kh.xiao@gmail.com (K.X.); william.laframboise@ahn.org (W.L.); david.bartlett@ahn.org (D.L.B.); ali.zaidi@ahn.org (A.H.Z.); 2Cleveland Clinic, Taussig Cancer Institute, Cleveland, OH 44106, USA; dadgarn@ccf.org; 3Beckman Research Institute of City of Hope, Monrovia, CA 91016, USA; ajgoel@coh.org

**Keywords:** appendiceal cancer, immune profiling, CD3 and CD8 T cells, prognostic biomarkers, tumor microenvironment, clinical outcomes

## Abstract

Cancers can often avoid detection by the immune system, influencing disease progression and shaping therapeutic approaches. Biomarkers like intra-tumoral lymphocyte density have provided prognostic insight in colorectal cancer, though little is known about how the immune system interacts with rare cancers like appendiceal cancer. Studying appendiceal cancer is often challenging due to its rarity, several subsets, and evolving classification. Advances in appendiceal cancer genomics have highlighted the need to study it, including microbiome and immune cell construct. This study focuses on how immune cells are distributed in appendiceal tumors, how this relates to tumor features like type and severity, and whether these immune factors can predict outcomes. CD3+ and CD8+ cells play a critical role in anti-tumor immunity. CD8+ T cells mediate direct cytotoxic responses, whereas CD3+ helper T cells coordinate immune modulation. These findings underscore the prognostic value of immune profiling in appendiceal cancer.

## 1. Introduction

Appendiceal cancer (AC) is a rare malignancy with an estimated incidence of less than 1% of all gastrointestinal cancers. The incidence of AC, around 1.2 cases per 100,000 [1], has been increasing over the past two decades, largely due to improved detection through advanced imaging and increased use of appendectomy for non-malignant conditions [2]. Despite its rarity, AC exhibits significant histologic and clinical heterogeneity, ranging from indolent mucinous neoplasms to aggressive high-grade adenocarcinomas [3]. Given this heterogeneity, understanding the immune microenvironment is crucial, as immune infiltration has been shown to influence tumor progression and response to therapy in other gastrointestinal malignancies [4,5]. However, compared to colorectal and gastric cancers, the immunologic characterization of AC remains limited. In this study, we aim to provide an initial characterization of the tumor immune microenvironment in AC by quantifying CD3+ and CD8+ lymphocyte densities and assessing their association with clinical and pathological variables.

Immune evasion is a central hallmark of cancer biology and a proven contributor to disease progression across many tumor types [6]. As such, it has become increasingly important to identify biomarkers defining the immunologic state of solid tumors, in order to gain insight into prognosis and select rational immunotherapeutic strategies for individual patients [7]. Among other candidate biomarkers of immunologic response or evasion in tumors, the measurement of intra-tumoral lymphocyte density has gained traction in colorectal cancer and other solid tumors, in which CD3+ and CD8+ T cell densities are quantified and utilized as prognostic and predictive assays [8,9,10]. For example, the Immunoscore^TM^ is an internationally validated clinical assay based on the separate calculation of CD3+ and CD8+ cell counts within tumors, followed by conversion of these counts to percentiles and averaging them together into a composite score. A high Immunoscore^TM^, indicative of a robust T cell infiltrate into a tumor, is a favorable prognostic marker now in common use in colorectal and other gastrointestinal cancers [4,5].

Despite widespread advancements in other malignancies, the immunologic characteristics of AC and pseudomyxoma peritonei (PMP) remain poorly understood. Research into the molecular characteristics of AC has been slowed by the rarity of the disease, the variety of histologic subtypes, and the constant evolution of classification systems over recent years [11]. In spite of these challenges, our understanding of the genomic landscape of AC has dramatically improved with the publication of several large mutational profiling series [11,12,13,14,15,16,17], and it is hoped that more comprehensive analyses of epigenetic and other molecular aspects of AC will follow closely behind [18]. As the molecular characterization of AC continues to unfold, it will be critical to concurrently study the AC tumor microenvironment, including the immunologic milieu within primary appendiceal tumors and the peritoneal cavity, as well as the microbiome of AC [19,20], as these features are likely to contain prognostic and predictive information far beyond what can be gained from studying tumor cells in isolation. It is unlikely that emerging immunotherapy options for other gastrointestinal malignancies will become available to patients with AC until a systematic characterization of this AC immune microenvironment has been achieved [21].

As a first step toward a broader immunologic characterization of AC, we sought to provide a descriptive analysis of the range of lymphocyte infiltration within AC. We hypothesized that, as in other gastrointestinal malignancies, these immune parameters may be associated with patient and tumor characteristics, specifically focusing on histologic subtype and grade, given the fundamental importance of these aspects of tumor biology in AC. Intra-tumoral lymphocytes play a critical role in anti-tumor immunity, both by mediating direct cytotoxic responses and by the secretion of cytokines that modulate the immunologic milieu and recruit additional cell types. Thus, we also sought to assess the prognostic impact of intra-tumoral CD3+ and CD8+ lymphocyte density on oncologic outcomes in patients with AC.

## 2. Material and Methods

### 2.1. Case Selection and Cohort Details

Archival tissue samples were compiled from a cohort of ninety-five unique patients diagnosed with appendiceal cancer (AC). Tissue samples from the primary appendiceal tumor were obtained whenever possible (*n* = 38 cases); in the remaining (*n* = 57) cases, peritoneal metastatic tissue was analyzed. Selection criteria included patients with histologically confirmed AC for whom archival tissue samples and comprehensive clinical follow-up data were available. When the primary tumor was available, the sample was used for analysis over peritoneal metastatic samples when metastatic specimens were collected during the same index operation. Demographic, clinical, and pathologic characteristics of the study group are presented in Table 1.

### 2.2. Compilation of Clinical Records and Clinicopathologic Features

Clinical records and pathologic features of the cohort were extracted from the Allegheny Health Network (AHN) electronic medical records system, including data on patient demographics (age and sex), clinical background (CEA level at time of diagnosis, treatment response, recent chemotherapy, history of cytoreductive surgery, Body Mass Index, and diabetes mellitus status), tumor characteristics (histologic diagnosis, grade, stage, primary versus metastatic disease status, presence of lymphovascular or perineural invasion, RAS mutation status, and microsatellite instability (MSI) status), and clinical outcomes (best response to therapy, date of progression, survival, and last follow up). Survival measures were calculated from the time of analytic tumor sample acquisition to the time of first clinical or radiographic evidence of disease progression or to the time of death. Censoring occurred at the time of last follow-up or at the time of dataset closure for final analysis (1 July 2024).

### 2.3. Tissue Handling and Acquisition

Formalin-fixed paraffin-embedded (FFPE) tissue blocks were retrieved from the archives of the AHN Pathology Institute, and selected slides were reviewed by a board-certified anatomic pathologist to confirm presence of neoplastic tissue for analysis. Tissue sections, including tumor and immediately adjacent non-neoplastic tissue, were cut at a thickness of 4 µm and mounted on charged slides for subsequent immunohistochemical staining.

### 2.4. Immunohistochemical Staining

Immunohistochemical (IHC) staining was performed to detect CD3+ and CD8+ T cells using the Leica Bond-Max automated biosystem (Leica Biosystems, Richmond, IL, USA). Tissue sections underwent pretreatment with heat-induced epitope retrieval using Epitope Retrieval 2 solution for twenty minutes (Leica Biosystems). Endogenous peroxidase activity was blocked using 3% hydrogen peroxide. Sections were subsequently incubated with ready-to-use primary antibodies: anti-CD3 (LN10, PA0553, Leica Biosystems) and anti-CD8 (4B11, PA0183, Leica Biosystems) for fifteen minutes. The detection was carried out using the Bond Polymer Refine Detection kit (DS9800, Leica Biosystems). After staining, the slides were dehydrated through a series of ethanol and xylene, followed by cover slipping. Examples of immunohistochemical stained slides are represented in Figure 1.

### 2.5. Digital Image Analysis and Cell Density Calculation

Digitally scanned images of the stained slides were analyzed using algorithms in the Leica Image Analysis Software within the eSlide Manager Spectrum Version 12.5.0.6145 (Leica Biosystems). Areas of interest were selected by a board-certified pathologist and were confirmed areas of tumor. The same field of view or region of interest (ROI) was used to assess both stained slides. CD3+ and CD8+ cell densities were quantified by counting positive cells within defined tumor areas, after manually tuning and validating the detection algorithm. Areas selected for analysis were neoplastic tissue and the immediate peri-tumoral stroma within 0.5 mm of neoplastic cells. Areas of acellular mucin were excluded from analysis. Cell density was defined as the number of positive cells per unit of analytic tissue area in each section (measured in square millimeters). CD3+ and CD8+ cell densities were then separately converted to percentile scores across all cases, and a composite metric (“I-score”) was derived by averaging these percentiles within each case in a manner analogous to the Immunoscore^TM^ [4]. The ratio of CD8+ to CD3+ cell density was also calculated and recorded for each sample.

### 2.6. Statistical Analysis

Descriptive statistics were calculated for demographic, clinical, and pathologic characteristics of the patient cohort, along with the densities of immune cells within the tumor samples. Association between immune parameters and clinicopathologic variables was tested using Spearman’s rho for continuous variables and the Kruskal–Wallis test for categorical variables. Survival outcomes (progression-free survival/PFS) were analyzed using Cox proportional hazards models, and subgroup comparisons were carried out using the log-rank test. All statistical analyses were conducted using STATA version 16 (StataCorp, College Station, TX, USA, 2019), and statistical significance was defined at an α error level of (*p*-value less than) 0.05.

## 3. Results

Tissue samples were evaluated from ninety-five unique patients in this cohort. Cohort descriptive statistics are presented in Table 1. The majority of the patients within the cohort had stage IV AC at time of resection (*n* = 69, 72.6%). Patients were divided into histologic subtypes as follows: low-grade appendiceal mucinous neoplasm (LAMN; *n* = 26, 27.3%), mucinous adenocarcinoma (mAC, *n* = 33, 34.7%), goblet cell carcinomas (gcAC, *n* = 15, 15.8%), signet ring cell carcinoma (srcAC, *n* = 9, 9.5%), or adenocarcinoma not otherwise specified (AC NOS; *n* = 12, 12.7%). The histologic grade was grade I in 35 (37.6%), grade 2 in 26 (28.0%), and grade III in 32 (34.4%) patients, respectively. Analyzed tumors were primary appendiceal lesions in 38 (40%) of cases and peritoneal metastatic lesions in the remaining 57 (60%). Thirty cases (50.8%) had documented nodal involvement at the time of analyzed tissue resection, nine (9.5%) cases had documented lymphovascular invasion (LVI), and fourteen (14.7%) cases had perineural invasion (PNI). Additional pathologic and clinical features, including carcinoembryonic antigen (CEA) level, extent of peritoneal disease (peritoneal carcinomatosis index/PCI), prior cytoreductive surgery, recent systemic therapy within 3 months prior to surgical resection, and best response to therapy are included in Table 1. Patients in this cohort underwent various treatments, including systemic chemotherapy, cytoreductive surgery, and immunotherapy, where applicable. The influence of these treatments on immune infiltration patterns was considered in our analysis. The total follow-up duration for the entire cohort was seven person-years, and the median progression-free and overall survival were 23 months and 37 months, respectively.

As expected, overall CD3+ cell density (median 397.9 cells/mm^2^; 95%CI [206.0, 756.9]) exceeded CD8+ cell density (median 225.6 cells/mm^2^; 95%CI [105.2, 415.1]). The distribution of CD3+ and CD8+ cell density by case was non-normally distributed across tumor samples, with evidence of rightward skew and kurtosis (*p* < 0.0001 for CD3+ and CD8+ lymphocyte density, Figure 2), reflecting the presence of a subset of extreme outliers for both high CD3+ and high CD8+ cell density. Both metrics exhibited wide variability (coefficient of variation, 90.3% and 116.6%, respectively). No difference in lymphocyte density was observed between appendiceal primary tumors vs. peritoneal metastases (Table 2). We assessed whether underlying patient factors were associated with CD3+ or CD8+ cell density and found an inverse association with age, whereas no difference in lymphocyte density or ratio was seen on the basis of sex (Table 2). On subgroup analysis, the effect of age appeared to be most prominent in patients above age 40, male patients, in primary vs. metastatic lesions, in mucinous AC vs. other histologic types, and in high- vs. low-grade tumors (Appendix A). Metabolic parameters, including body mass index (BMI), history of diabetes mellitus, and hemoglobin A1c level were unassociated with CD3+ and CD8+ lymphocyte density.

Given the dominant impact of histologic subtype and grade on disease biology in AC, we next examined whether these variables are associated with lymphocyte density in tumor tissue. Among histologic tumor types, LAMN contained the highest density of both CD3+ and CD8+ lymphocytes as well as I-score, which is a derivative of CD3+ and CD8+ counts performed in a manner analogous to the Immunoscore^TM^ (*p* = 0.008 for CD3+ density and *p* = 0.03 for IS; Table 2/Figure 3). A detailed analysis of lymphocyte density by histologic subtype, presented in Appendix A, confirms the association of lower CD3+ and CD8+ lymphocyte densities and I-score in both mucinous and non-mucinous AC relative to LAMN. We also found a strong association of higher CD3+ and CD8+ cell density with low histologic grade (Table 2, Figure 4). I-score was likewise strongly associated with grade. These associations remained significant in the subset of invasive AC, i.e., when excluding LAMN. Taken together, the findings suggest substantial immunologic diversity across the spectrum of histologic subtype and grade among AC.

We then tested for association between tumor volume and lymphocyte density, using AJCC stage, PCI, and CEA level as surrogate markers for disease extent. No association between AJCC stage and lymphocyte counts was found across the cohort, but when excluding LAMN, we found association of CD3+, CD8+, and IS with AJCC stage (*p* = 0.04, *p* = 0.06, *p* = 0.04, respectively). Although PCI was not associated with any cell density or ratio metric when analyzed across the entire cohort (Table 3), we found evidence on subgroup analysis of negative association between PCI and CD8+ lymphocyte density in LAMN (ρ = −0.5, *p* = 0.04) and AC-NOS (ρ = −0.7, *p* = 0.07), as opposed to a positive association between these variables in signet ring cell AC (ρ = 0.9, *p* = 0.05; Appendix A). CEA was associated with lymphocyte density when restricting the analysis to patients with values above the threshold of quantitative detection (>1.8 ng/dL, Table 3). No association was seen between lymphocyte density nodal status, presence of LVI or PNI, RAS, or MSI status. In terms of prior therapy, we saw no association of lymphocyte density parameters with previous CRS or recent chemotherapy within three months of tissue acquisition.

The distribution characteristics of lymphocyte density in AC imply the existence of a set of outliers, which in our series were defined as tumors with >1000 CD3+ (*n* = 15) or >500 CD8+ (*n* = 20) lymphocytes, respectively. High-CD3+ outliers were more likely to have LAMN (60% vs. 22%, *p* = 0.005) and low histologic grade (60% vs. 33.3%, *p* = 0.05), with elevated CEA levels (median 9.1 ng/mL, IQR [1.5, 61] vs. 3.9 [2.3, 11.5]; *p* = 0.02). High-CD8+ outliers were associated with lower age (median 49 years [40, 60] vs. 58 [41, 66]; *p* = 0.02), mucinous histologic subtypes (LAMN or mAC), low-grade morphology (55% vs. 33%, *p* = 0.06) and elevated CEA levels (median 6.3 ng/mL, IQR [1.5, 61] vs. 3.9 [2.3, 11.5]; *p* = 0.02).

Lymphocyte density and ratio measures were then evaluated for association with oncologic outcome including best response and survival. Best response to systemic therapy—when considered as a continuum from progression (PD) to partial response (PR) to stable disease (SD) to no evidence of disease (NED)—was associated with increased CD8+ lymphocyte density and I-score (Table 2). Analysis of overall survival was omitted, since only fourteen patients died during follow-up, precluding statistical power for subgroup comparisons. Sixty-three patients (66.3%) experienced progression events during follow-up. Lymphocyte density and I-score were next assessed individually as continuous variables for association with PFS, and then as dichotomous variables using stratification cutoffs. For CD3+ and CD8+ cell density and I-score, the 70th percentile was selected as on univariable analysis, PFS was not associated with CD3+ or CD8+ cell densities or I-score but was associated with CD8+:CD3+ density ratio (HR 0.6, 95%CI [0.4, 1.0]); *p* = 0.04; Table 4, Figure 1). Given the dominant effect of histologic grade on survival in AC, along with the potentially confounding effects of patient age, tumor volume (PCI) and tumor grade, multivariable analysis was performed, confirming that the prognostic significance of the CD8+:CD3+ cell density ratio was independent of age, grade and site of tissue biopsy (HR 0.39, 95%CI [0.20, 0.73], *p* = 0.004; Table 5/Figure 5).

Finally, a dichotomous variable based on CD8+:CD3+ density ratio was created using the Cutoff Finder application to determine a stratification “cutoff” value (https://molpathoheidelberg.shinyapps.io/CutoffFinder_v1/, accessed on 15 October 2024) as previously described [22]. This dichotomous variable, with an optimal cutoff determined to be 0.9, was confirmed to be significantly associated with prolonged PFS on univariable analysis and on multivariable analysis incorporating patient age, tumor grade, PCI, and site of tumor biopsy (Table 3 and Table 4). A discriminatory cutoff based on underlying CD3+ or CD8+ cell density could not be identified in this dataset, although PFS in low grade AC was associated with tumor I-score > 70th percentile (HR 0.4, 95%CI [0.2, 1.0], *p* = 0.04; Table 4 and Appendix A).

## 4. Discussion

The immune microenvironment within solid tumors is increasingly recognized as a vital indicator of host response to neoplastic growth, and as a key biomarker of prognosis and predictor of immunotherapy efficacy. Specific lymphocyte subsets—including cytotoxic CD8+ T cells, memory T cells, T_H_1 cells, T_FH_ cells and B cells—are indicative of an adaptive immune response and favorable prognosis, whereas the presence of regulatory T cells (T_REG_) and myeloid cells is suggestive of a maladaptive response and poor prognosis [8]. While these trends are consistent across diverse primary tumor types, including colorectal and other digestive cancers, there has been little cohesive data on immunoprofiling of appendiceal cancer to date, as a result of the rarity of this disease and the intensive technical demands of tissue handling and characterization. In this study, we report an initial characterization of lymphocyte density in AC, as a first step toward a comprehensive description of the immune microenvironment within this diverse group of tumors. Our findings provide a foundation for further research into the immunologic heterogeneity of AC, including the classification of tumors into ‘hot’ (immune-infiltrated) and ‘cold’ (immune-excluded or immune-deserted) phenotypes [23].

The appendix, in immunologic terms, is a unique organ with features distinct from the small intestine or colon, as previously reviewed [24,25]. The appendix differs from the colon in containing abundant lymphoid follicles in the submucosa and lamina propria, with an overlying dome epithelium marked by increased intra-epithelial lymphocytes and antigen-presenting cell constituents [24]. The functional significance of the appendiceal immunologic apparatus is believed, by analogy to Peyer’s patches in the small intestine, to be closely linked to maintenance of gut immunity and tolerance of commensal organisms. The lymphoid follicles in the appendix express CCL21 in high endothelial venules and parafollicular areas, contributing to the recruitment of B and T cells to the local immune environment and the return of activated dendritic cells to the regional lymph nodes [26]. M cells within the dome epithelium are important in antigen sampling and presentation via transcytosis, and the appendix is a dominant site of IgA production in the gut [27,28], contributing substantially to its dense biofilm, which decreases sequentially with distal progression through the colon [29]. Due to its dense biofilm and relative protection from fecal flow and disturbances in the setting of diarrheal illness or other pathology, the appendix is believed by some authors to constitute a ‘safe house’ harboring an inoculum of commensal organisms that maintain overall stability in the gut microbiome [30]. While it is clear that the unique immunologic niche within the normal appendix differs relative to other anatomic sites, it remains unknown to what degree this dynamic immunologic milieu might function to promote carcinogenesis and tumor progression, or influence response.

Intra-epithelial lymphocytes in the normal appendix are primarily T cells, which increase in number in the setting of luminal infection, especially parasitic infections [31]. In this study, we demonstrated that CD3+ and CD8+ cell densities vary widely among AC specimens, with a subset of outlier tumors exhibiting extremely dense lymphocyte populations. These outliers were disproportionately—but not exclusively—younger patients with low-grade, mucinous tumor types and elevated CEA levels. Additional clinical, pathologic, genetic, or other factors may explain the propensity for lymphocyte infiltration seen within this group of cases (BRAF mutation or microsatellite instability, for example) [32], but further definition of any underlying mechanistic relationships will require follow-up studies with larger datasets. As in CRC, patients with AC in this outlier group may prove ideal candidates for investigational use of immunotherapy [7], for whom surgery and conventional cytotoxic agents have proven ineffective at controlling disease progression.

We found a strong association between lymphocyte density and grade, with decreasing density linked to progressively higher grade. Similarly, we found differences in tumor lymphocyte content among histologic tumor types, with an overall more robust infiltrate in mucinous tumors versus non-mucinous tumors. The reasons for these associations are unclear. On the one hand, a higher mutational burden and neo-antigen level might be expected in higher grade tumors, in which case a less robust immune response would seem counterintuitive. However, high-grade tumors often exhibit an immunosuppressive phenotype in which chronic fibrosis is favored over an adaptive immunologic response. Low grade tumors, by contrast, are often accompanied by extensive mucin deposition, which may create local inflammation and subsequent recruitment of a prominent lymphocytic infiltrate. In colorectal cancer, mucinous subtypes, including mucinous adenocarcinoma and mucin-rich signet cell variants, are notorious for exuberant lymphocytic response, sometimes described as “Crohn’s-like”, and it is possible that similar patterns in AC may be elucidated in future studies [33]. Further exploration of the immune contexture in greater numbers of clinically significant AC subtypes, such as mucinous, signet ring cell or goblet cell adenocarcinomas and neuroendocrine tumors will be an important next step.

We also found strong associations between patient age and lymphocyte density within AC. Interestingly, this effect appeared to be accentuated within primary tumors from male patients with high-grade mucinous appendiceal AC. In the normal appendix, the overall density of lymphoid tissues, including follicles and germinal centers, as well as absolute and relative B and T cell densities are known to evolve over the human lifespan, expanding and peaking during childhood, with progressive decline with aging [34]. How the evolution of the background appendiceal milieu over the lifespan impacts the development and progression of AC remains to be seen, although our results are certainly consistent with an age-related influence. Mixed results have been reported regarding the impact of age on lymphocytic infiltration in CRC, with large studies in western and Asian populations indicating no association with age [35,36], and other studies suggesting an interaction between age and lymphocyte density in prognosis and prediction of response to systemic chemotherapy [10,35].

Our study has several important limitations. The immune contexture of solid tumors is a complex endpoint of tumor- and patient-related input variables, including genetic and epigenetic factors both from the germline and tumor, host metabolic parameters, and microbiome influence. The tumor microenvironment can change over time and, given the retrospective design and reliance on archived tissue samples, we may not have captured the dynamic nature of immune responses, particularly changes that may occur during or after treatment. Although we addressed this point by ruling out differences among patients with or without recent chemotherapy and prior CRS, future prospective or longitudinal studies will be required to yield definitive data on this question. Similarly, while we focused predominantly on CD3+ and CD8+ lymphocytes given the wide acceptance of the Immunoscore^TM^ as a baseline immunologic ‘thermometer’ among solid tumors [37], our analysis may be overly simplistic, neglecting additional immune cell subsets and mediators that could play a central role within the tumor microenvironment (TME). Further delineation of lymphocyte subsets—including activated, exhausted, regulatory, or memory cells—may shed additional light on the nature of the immune infiltrate and its relationship to tumor grade, which is the single-most important prognostic indicator in AC. Inter-observer variability is always a concern in estimating cell counts within histologic slides, and while our use of digital image analytic software should mitigate specimen-to-specimen counting variation in our dataset, our interpretations would benefit from independent validation in separate tumor cohorts. Moreover, while our overall findings in primary and metastatic tumors were not significantly different, our study may have been underpowered to identify subtle differences across these environments, and any confounding effects of tumor location will need to be carefully untangled and supplemented with ex vivo models to study the difference between the primary tumor and peritoneal metastatic immune microenvironments [38].

Notwithstanding these limitations, it bears repeating that a fairly rudimentary immune microenvironment parameter—namely the intra-tumoral CD8+:CD3+ lymphocyte ratio—was prognostically independent of histologic grade and tumor volume (PCI), which are two of the most consistently identified variables driving the prognosis of AC across decades of clinical research. This finding is reminiscent of the outperformance of conventional tumor-node-metastasis (TNM) staging in colorectal cancer by the Immunoscore^TM^ [39]. In many cancers, there is intense ongoing interest in defining robust biomarkers to characterize the tumor immune microenvironment, and to predict response to immune checkpoint inhibitor therapy and other evolving treatment modalities. The microenvironment is known to vary from tumors of one organ to another, and to be heterogeneous even within an individual tumor, making it very challenging to envision a single biomarker applicable to all clinical scenarios [37]. AC contains a diverse group of histologic subtypes, and it is very likely that the immune contexture will differ among them, reinforcing the importance of additional studies with more sophisticated measures of cellular and molecular constituents of the AC TME. Much work remains in assessing the clinical applicability of lymphocyte quantification in AC, particularly in the context of patient stratification and therapeutic decision-making. Prospective studies integrating this biomarker into risk models and evaluating its predictive value in immunotherapy response are needed to fully establish its translational relevance.

Previous studies highlighting the immune microenvironment of AC are generally scarce, but we did find a recent histologic analysis in which peri-mucinous immune aggregates were found to be a feature of low-grade AC, while intra-tumoral immune aggregates were noted to be more common in high grade tumors [40]. Although only a small number of cases were analyzed, and the findings were based solely on histology without immunohistochemical studies, these results corroborate our overall impression that the immune microenvironment may differ in important ways based on histologic grade, and it remains to be seen to what degree a more active immune response contributes to the comparatively indolent biology and improved prognosis in patients with low-grade relative to high-grade AC. Importantly, our findings are also corroborated by preliminary findings from a scRNA-seq dataset acquired from peritoneal metastases in 11 cases of AC [41]. The authors reported substantial heterogeneity among histologic subsets of AC, with CD8+ T cell abundance appearing highest in LAMN and low-grade mAC, similar to our findings in the present series. Additionally, the authors reported upregulation of cytotoxic and cancer-progression-associated genes in T cells from tumors, as well as an influx of plasma cells and other effector cell types, myeloid cell subsets and CAFs in peritoneal deposits from AC. These results support the continued effort to characterize the immune contexture within AC as a potential source of targeted interventions.

## 5. Conclusions

As in other solid tumors, the TME in AC is an important potential source of prognostic and predictive information. Future, more comprehensive studies of the immune landscape in AC are planned and will need to include larger tumor sample sizes and broader sets of immune biomarkers, such as additional T and B cell subsets, myeloid cells, and expression of immune checkpoint molecules. Longitudinal studies that monitor immune profiles before and after therapeutic intervention could offer valuable insights into the temporal dynamics of immune responses and their predictive value for efficacy. In AC, the primary tumor within an individual patient exists in the broader context of the patient’s overall mucosa-associated lymphoid tissue (MALT), peritoneal immune environment, and systemic immune milieu [42]. Dissecting the relative contributions of these overlapping layers on AC tumor biology will be a complex task, but by integrating immune profiling with other emerging molecular and genetic data, we hope to identify subsets of AC patients in whom immunomodulatory or combination regimens may enhance survival and improve quality of life.

## Figures and Tables

**Figure 1 cancers-17-00542-f001:**
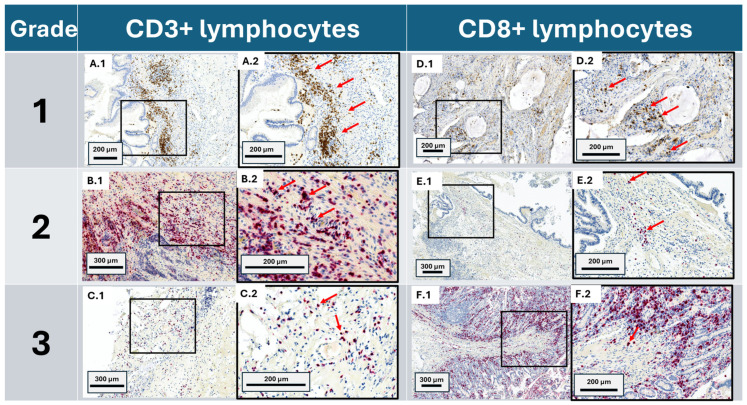
Immunohistochemical intra-tumoral lymphocyte staining of appendiceal cancer in various grades. CD3+ lymphocytes and CD8+ lymphocytes are represented on the left and right sides, respectively. (**A.1**) (with magnified field in (**A.2**)) represents a grade 1 LAMN with high CD3+ density within the submucosa and lamina propria (889.0 cells/mm^2^). (**B.1**) (with magnified field in (**B.2**)) represents a grade 2 AC NOS with intermediate CD3+ density within the lamina propria (568.7 cells/mm^2^). (**C.1**) (with magnified field in (**C.2**)) represents a grade 3 mAC with low CD3+ density within the submucosa and lamina propria (442.2 cells/mm^2^). (**D.1**) (with magnified field in (**D.2**)) represents a grade 1 mAC with high CD8+ density within the lamina propria (1366.4 cells/mm^2^) surrounding globules of acellular mucin for which the area of was excluded in our analysis of the region of interest. (**E.1**) (with magnified field in (**E.2**)) represents a grade 2 LAMN with intermediate CD3+ density within the lamina propria (316.4 cells/mm^2^). (**F.1**) (with magnified field in (**F.2**)) represents a grade 3 mAC with low CD3+ density within the submucosa (25.8 cells/mm^2^) as well as high lymphocytic densities within the mucosa, which was excluded in our analysis of intra-tumoral cellular densities. Red arrows depict examples of and are adjacent to either chromogen red or brown positive lymphocytes which were selected for in our analysis.

**Figure 2 cancers-17-00542-f002:**
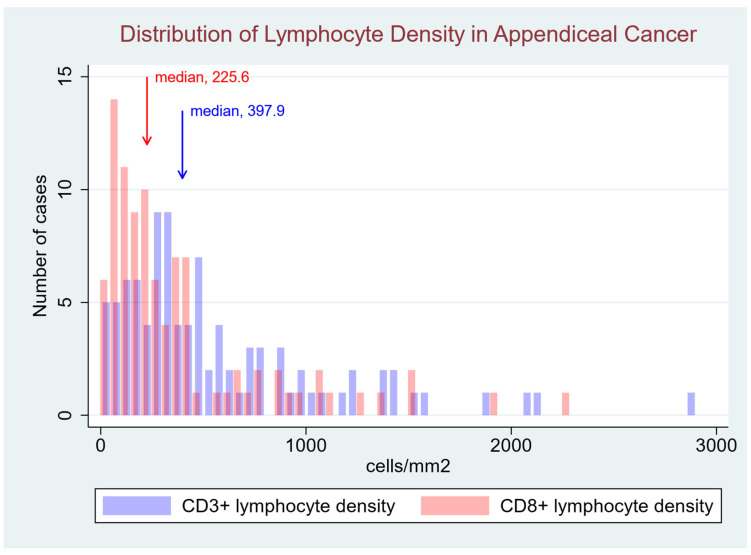
Distribution of Lymphocyte Density in Appendiceal Cancer. This histogram represents the distribution of **CD3+** (purple) and **CD8+** (red) lymphocyte densities across the cohort. The overall CD3+ cell density (median 397.9 cells/mm^2^; 95% CI [206.0, 756.9]) exceeded the CD8+ cell density (median 225.6 cells/mm^2^; 95% CI [105.2, 415.1]). The distribution was non-normal, exhibiting rightward skew and kurtosis (*p* < 0.0001 for both CD3+ and CD8+ densities). The purple bars represent the density of CD3+ T cells, which include all T lymphocytes, whereas the red bars represent the subset of CD8+ cytotoxic T cells, which inherently express CD3 as part of the T cell receptor complex.

**Figure 3 cancers-17-00542-f003:**
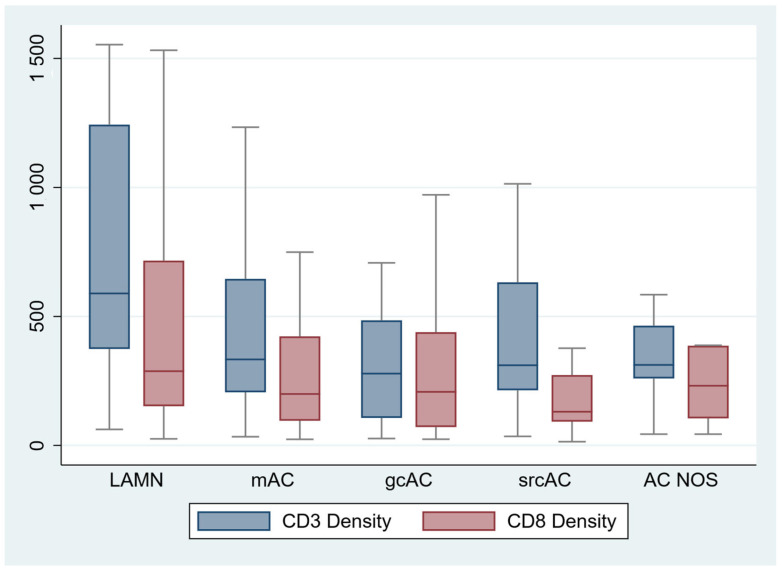
Comparison of four distinct and a subclassified histotype group including low grade appendiceal mucinous neoplasm (LAMN), mucinous adenocarcinoma (mAC), goblet cell adenocarcinoma (gcAC), signet ring cell adenocarcinoma (srcAC), and adenocarcinoma not otherwise specified (AC NOS). Among histologic tumor types, LAMN contained the highest density (589.3 [374.0, 1243.2]) of both CD3+ (*p* = 0.02). There was no statistical difference in CD8 densities, although LAMN had the highest concentrations (287.9 [152.2, 716.3], *p* < 0.08).

**Figure 4 cancers-17-00542-f004:**
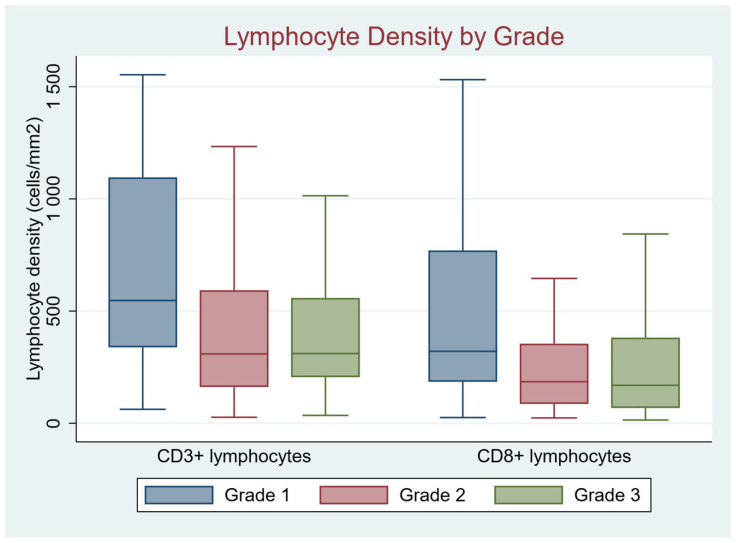
In low histologic grade, there was a strong association with higher CD3+ density (547.2 [338.6, 1095.8] compared to 309.2 [161.7, 593.0] and 310.9 [205.3, 558.6] for Grade 2 and Grade 3, respectively, *p* < 0.02). Similarly, CD8+ density was also higher in low grade tumors (320.7 [184.8, 770.3] compared to 185.3 [86.3, 354.5] and 169.3 [68.2, 381.7] for Grade 2 and Grade 3, respectively, *p* < 0.02).

**Figure 5 cancers-17-00542-f005:**
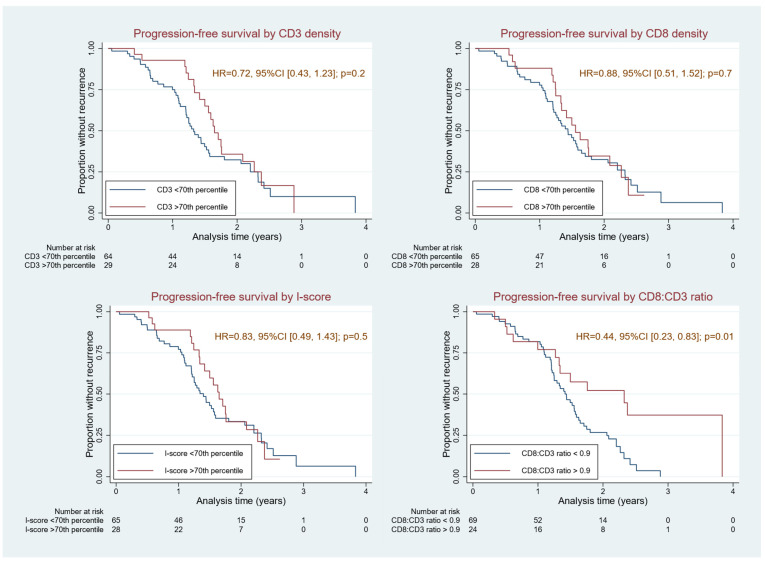
On univariable analysis, PFS was not associated with CD3+ or CD8+ cell densities or I-score but was associated with CD8+:CD3+ density ratio (HR 0.6, 95%CI [0.4, 1.0]); *p* = 0.04). For CD3+ and CD8+ cell density and I-score, the 70th percentile was selected as a cutoff by convention. On multivariable analysis, CD8+:CD3+ cell density ratio was independent of age, grade, and site of tissue biopsy (HR 0.39, 95%CI [0.20, 0.73], *p* = 0.004) and a predictor of improved PFS in appendix cancer.

**Table 1 cancers-17-00542-t001:** Patient characteristics.

Characteristic	Number of Patients	Median (Q1, Q3)	Standard Deviation	Percentage
	*n* = 95			
Age at diagnosis		57 (48, 66)	12.4	
Sex	Female = 41			43.2%
	Male = 54			56.8%
CEA (ng/mL)	*n* = 76	3.9 (2, 14.8)	183.8	
Histology	LAMN = 26			27.3%
	Adenocarcinoma = 10			10.5%
	Mucinous adenocarcinoma = 33			34.7%
	Goblet cell carcinoma = 15			15.8%
	Signet ring cell carcinoma = 9			9.5%
	Sessile serrated lesion = 1			1.1%
	Villous adenoma = 1			1.1%
Primary tumor Metastatic peritoneal lesion	*n* = 38 *n* = 57			40% 60%
Primary tumor with dissemination	*n* = 21			55.3%
Grade	1 = 35			37.6%
	2 = 26			28%
	3 = 32			34.4%
Stage	0 = 8			8.4%
	1 = 0			0%
	2 = 14			14.8%
	3 = 4			4.2%
	4 = 69			72.6%
Nodal Status	*n* = 61			Negative = 50.8%
				Positive = 49.2%
Lymphovascular invasion (in primary tumors)	*n* = 35			Identified = 25.7% (*n* = 9, 9.5% of all)
Perineural invasion (in primary tumors)	*n* = 29			Identified = 48.3% (*n* = 14, 14.7% of all)
RAS mutation status	Wild type (*n* = 6)			6.3%
	KRAS Mutation (*n* = 4)			4.2%
MSI status	Stable (*n* = 37)			39%
	High (*n* = 2)			2.1%
Peritoneal Cancer Index (PCI)		26 (18, 33)	9.1	
Completeness of Cytoreduction (CC) of those underwent CRS (*n* = 68)	CC0 (*n* = 36)			52.9%
	CC1 (*n* = 29)			42.7%
	CC2 (*n* = 1)			1.5%
	CC3 (*n* = 2)			2.9%
Previous CRS	*n* = 39			41.1%
Chemotherapy within 3 months of tissue acquirement	Yes (*n* = 19)			20%
Best response to therapy	NED (*n* = 23)			24.2%
	PR (*n* = 32)			33.7%
	Stable disease (*n* = 8)			8.4%
	Progression (*n* = 32)			33.7%
Progression noted at any time during follow up	*n* = 63			66.3%
Median progression free survival (months)		23		
Mortality	*n* = 14			14.7%

**Table 2 cancers-17-00542-t002:** Association of lymphocyte density and combination metrics with clinical and pathologic characteristics in appendiceal cancer. Cellular densities are expressed per mm^2^ of examined tissue. *p* values shown are for Kruskal–Wallis test of equivalence among subsets by grade, sex, or primary vs. metastatic tumor location.

	CD3 Density		CD8 Density		I-Score		CD8:CD3 Ratio	
	Median (IQR)	*p*	Median (IQR)	*p*	Median (IQR)	*p*	Median (IQR)	*p*
TOTAL	397.9 (206.0, 756.9)		225.6 (105.2, 415.1)		48.9 (27.6, 72.3)		0.6 (0.4, 1.0)	
FEMALE	365.8 (171.1, 645.7)	0.4	189.2 (113.7, 348.8)	0.4	37.7 (28.2, 66.5)	0.3	0.6 (0.4, 0.9)	0.9
MALE	447.4 (259.8, 773.4)		273.1 (92.2, 439.1)		53.1 (26.6, 73.9)		0.6 (0.4, 1.0)	
PRIMARY	333.5 (204.7, 769.5)	0.9	222.7 (92.2, 460.3)	1.0	51.6 (27.6, 73.4)	0.9	0.6 (0.4, 1.0)	0.7
METASTATIC	419.2 (214.1, 707.8)		225.6 (105.2, 415.1)		48.4 (28.2, 71.2)		0.6 (0.4, 0.9)	
LAMN	589.3 (374.0, 1243.1)	0.02	287.9 (152.2, 716.3)	0.06	62.7 (34, 87.7)	0.02	0.6 (0.4, 0.7)	0.8
Mucinous adenocarcinoma	333.5 (206.0, 645.7)		199.4 (95.7, 422.8)		48.4 (31.2, 66.5)		0.6 (0.5, 1.1)	
Goblet cell carcinoma	278.6 (106.7, 484.9)		207.8 (71.3, 439.1)		37.7 (26.6, 57.9)		0.6 (0.4, 1.6)	
Signet ring cell adenocarcinoma	310.8 (214.1, 632.3)		130.6 (92.2, 272.8)		32.4 (28.2, 68.6)		0.4 (0.3, 0.5)	
Adenocarcinoma NOS	312.2 (259.7, 464.2)		231.4 (105.0, 386.5)		47.0 (23.8, 59)		0.8 (0.5, 1.1)	
AJCC Stage 0	705.4 (457.9, 1107.2)	0.9	346.4 (208.8, 615.3)	0.7	65.9 (50.3, 84.5)	1.0	0.5 (0.4, 0.6)	0.2
Stage II	369.9 (136.4, 712.2)		212.8 (86.3, 316.4)		41.5 (26.6, 57.9)		0.5 (0.3, 0.9)	
Stage III	88.5 (40.5, 331.9)		64.5 (19.3, 152.3)		11.4 (2.1, 36.6)		0.5 (0.4, 0.7)	
Stage IV	342.1 (258.5, 756.9)		225.6 (109.9, 439.1)		48.9 (29.8, 72.3)		0.6 (0.4, 1.1)	
Grade 1	547.2 (338.6, 1095.8)	0.02	320.7 (184.8, 770.3)	0.009	57.9 (37.2, 87.7)	0.01	0.6 (0.4, 1.0)	0.08
Grade 2	309.2 (161.7, 593.0)		185.3 (86.3, 354.5)		38.8 (24.4, 55.8)		0.5 (0.5, 1.1)	
Grade 3	310.9 (205.3, 558.6)		169.3 (68.2, 381.7)		41.7 (23.6, 61.9)		0.5 (0.3, 0.9)	
Nodal involvement	352.4 (214.1, 769.5)	0.3	200.4 (92.2, 422.8)	0.7	47.8 (24.4, 73.9)	0.9	0.5 (0.3, 1.0)	0.07
Negative nodal status	333.5 (162.0, 484.9)		232.2 (128.0, 376.9)		45.7 (30.9, 57.9)		0.7 (0.5, 1.1)	
LVI present	391.6 (106.7, 707.8)	0.4	207.8 (149.4, 316.4)	0.5	40.4 (26.6, 70.7)	0.4	0.6 (0.4, 1.7)	0.7
Without LVI	445.3 (324.0, 889.0)		280.1 (152.2, 636.0)		53.7 (34, 80.3)		0.6 (0.4, 1.0)	
PNI present	333.1 (171.1, 482.9)	0.3	266.1 (71.3, 439.1)	0.7	50.8 (29.8, 55.3)	0.7	0.6 (0.4, 1.6)	0.4
Without PNI	448.3 (338.6, 889.0)		2322.2 (86.3, 3545)		52.1 (34, 72.3)		0.5 (0.3, 0.6)	
Recent chemotherapy	310.8 (233.8, 632.2)	0.5	213.1 (130.6, 439.1)	0.9	44.1 (35.6, 55.8)	0.8	0.6 (0.4, 1.6)	0.4
No chemotherapy within 3 months	440.9 (198.8, 875.2)		238.6 (101.7, 428.1)		50.3 (27.6, 73.6)		0.6 (0.4, 0.9)	
Prior CRS	365.8 (259.8, 773.4)	0.7	273.5 (109.9, 438.6)	0.6	48.9 (31.2, 72.3)	0.6	0.6 (0.4, 0.9)	0.6
No prior CRS	401.4 (180.0, 686.0)		213.2 (89.2, 419.7)		49.2 (26.6, 72.8)		0.7 (0.4, 1.1)	
Response to treatment: NED	442.2 (162.0, 971.2)	0.2	300.9 (181.3, 460.3)	0.02	55.8 (27.6, 80.8)	0.05	0.6 (0.5, 1.0)	0.2
Response to treatment: PR	352.2 (276.7, 771.5)		280.7 (115.8, 680.9)		50.8 (33.5, 77.9)		0.6 (0.4, 1.3)	
Response to treatment: Stable disease	386.4 (155.7, 792.6)		305.1 (125.8, 787.6)		51.0 (25.7, 80.3)		0.8 (0.6, 1.5)	
Response to treatment: Progression	358.0 (146.3, 670.0)		174.4 (74.6, 294.6)		42.8 (20.4, 62.7)		0.5 (0.3, 0.7)	

**Table 3 cancers-17-00542-t003:** Lymphocyte density and combination metrics are associated with patient age, but not carcinoembryonic antigen (CEA) or peritoneal carcinomatosis index (PCI) in appendiceal cancer. *p* values shown for Spearman’s rank correlation coefficient (rho).

Association of Lymphocyte Density with Continuous Variables						
	Age		PCI		CEA	
	Rho	*p*	Rho	*p*	Rho	*p*
CD3 density	−0.2	0.07	0.04	0.8	0.2	0.2
CD8 density	−0.2	0.04	−0.05	0.7	0.02	0.9
I-score	−0.2	0.04	−0.01	0.9	0.1	0.4
CD8:CD3 ratio	−0.04	0.7	−0.1	0.3	−0.2	0.1

**Table 4 cancers-17-00542-t004:** Association of lymphocyte density and combination metrics with progression-free survival in appendix cancer. *p* values are derived from log-rank tests, which were performed on all patients (center-left columns), followed by analyses stratified by grade (middle columns) and primary vs. metastatic tumors (right columns). HR, hazard ratio; CI, confidence interval.

Variable	Cutoff	All AC			Low-Grade AC			Intermediate/High Grade AC			Primary AC			Metastatic (Peritoneal) AC		
		HR	95% CI	*p*	HR	95% CI	*p*	HR	95% CI	*p*	HR	95% CI	*p*	HR	95% CI	*p*
CD3+ cell density	continuous	1.0	[0.9–1.0]	0.9	1.0	[1.0–1.0]	0.5	1.0	[1.0–1.0]	1.0	1.0	[1.0–1.0]	0.04	1.0	[1.0–1.0]	0.6
CD8+ cell density	continuous	1.0	[1.0–1.0]	0.3	1.0	[1.0–1.0]	0.07	1.0	[1.0–1.0]	0.7	1.0	[1.0–1.0]	0.07	1.0	[1.0–1.0]	0.8
I-score	continuous	1.0	[0.9–1.0]	0.9	1.0	[1.0–1.0]	0.3	1.0	[1.0–1.0]	1.0	1.0	[1.0–1.0]	0.4	1.0	[1.0–1.0]	0.7
	>70th percentile	0.8	[0.5–1.4]	0.5	0.4	[0.2–1.0]	0.04	1.0	[0.5–2.3]	0.9	0.4	[0.2–1.0]	0.05	1.4	[0.7–2.8]	0.4
CD8:CD3 cell ratio	continuous	0.6	[0.4–1.0]	0.04	0.4	[0.2–0.8]	0.01	0.7	[0.4–1.2]	0.2	0.6	[0.3–1.2]	0.2	0.6	[0.3–1.2]	0.1
	>0.9	0.4	[0.2–0.8]	0.01	0.4	[0.1–1.0]	0.04	0.5	[0.2–1.1]	0.08	0.3	[0.1–1.1]	0,07	0.5	[0.2–1.2]	0.1

**Table 5 cancers-17-00542-t005:** CD8:CD3 ratio is an age- and grade-independent predictor of improved progression-free survival in appendix cancer. The analysis was performed with CD8:CD3 ratio treated as a continuous variable and as a dichotomous variable at a cutoff point of 0.9. CI, confidence interval; PCI, peritoneal carcinomatosis index.

Multivariable Analysis: Progression-Free Survival			
	Hazard Ratio	95% CI	Log-Rank *p*
CD8:CD3 ratio (continuous)	0.39	[0.20, 0.73]	0.004
Age	1.00	[0.98, 1.03]	0.88
Grade			
I	1.32	[0.67, 2.6]	0.81
II	0.64	[0.30, 1.37]	0.25
III	0.59	[0.266, 1.29]	0.19
PCI	0.98	[0.95, 1.02]	0.39
CD8:CD3 ratio > 0.9	0.32	[0.14, 0.72]	0.006
Age	1.00	[0.97, 1.02]	0.79
Grade			
I	1.45	[0.74, 2.88]	0.28
II	0.72	[0.34, 1.51]	0.39
III	0.78	[0.36, 1.65]	0.51
PCI	0.98	[0.94, 1.02]	0.25

## Data Availability

The raw data supporting the conclusions of this article will be made available by the authors on request.

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
