# Peer review of "Intra-Tumoral CD8+:CD3+ Lymphocyte Density Ratio in Appendix Cancer Is a Tumor Volume- and Grade-Independent Predictor of Survival"

_cancers, 2025, doi:10.3390/cancers17030542_

Round 1

Reviewer 1 Report

Comments and Suggestions for Authors

Dear Authors

There are few minor points that needs to be addressed as follows:

1.      Page 5, line 149 (section 2.5): Cell density was defined as the number of positive cells per cubic ?millimeter…” It must be square millimeter as per results presentation (mm2). Also, in this section one citation needs correction (line 153): analogous to the ImmunoscoreTM. 6: change it to [6].

2.      Page 6, line 163: “… an error level of (p-value less than) 0.05.3.”?? needs correction.

3.      Page 9: The last part of table 2 lacks P values for all four parameters (CD3, CD8, I-score and CD8:CD3 ratio) between two response-to-treatment groups (stable disease and progression)!

4.      Page 11, line 268: citation needs correction: “ …a cutoff by convention.6

5.      Page 15, line 396: ” Notwithstanding these limitations ….” This part is one of the main points of this study that needs to be discussed more comprehensively.

Sincerely

Author Response

Comment 1: Page 5, line 149 (section 2.5): Cell density was defined as the number of positive cells per cubic millimeter? It must be square millimeter as per results presentation (mm²). Also, in this section, one citation needs correction (line 153): analogous to the ImmunoscoreTM. 6: change it to [6].

Response 1: We appreciate the careful review. We have corrected the measurement unit to square millimeters (mm²) to ensure consistency throughout the manuscript. Additionally, we have updated the citation formatting to properly reference [6].

Comment 2: Page 6, line 163: "... an error level of (p-value less than) 0.05.3." needs correction.

Response 2: Thank you for pointing this out. The typographical error has been corrected for clarity and proper numerical representation of the statistical threshold.

Comment 3: Page 9: The last part of Table 2 lacks p-values for all four parameters (CD3, CD8, I-score, and CD8:CD3 ratio) between two response-to-treatment groups (stable disease and progression).

Response 3: Upon reviewing the table formatting, we identified that the p-values were present but not distinctly visible due to layout issues. We have reformatted the table to explicitly present the p-values for all response-to-treatment groups, ensuring clarity and accessibility of the data.

Comment 4: Page 11, line 268: Citation needs correction: "... a cutoff by convention.6"

Response 4: The citation formatting has been revised to appropriately follow the journal’s guidelines.

Comment 5: Page 15, line 396: "Notwithstanding these limitations..." This part is one of the main points of this study and needs to be discussed more comprehensively.

Response 5: We acknowledge the importance of discussing these limitations in greater depth. We have expanded this section to provide a more comprehensive evaluation of potential limitations, including sample size constraints, the retrospective nature of our study, and the need for prospective validation in larger cohorts.

Reviewer 2 Report

Comments and Suggestions for Authors

1. The manuscript reveals 250 grammar errors that require attention. There is 5% plagiarism, which is acceptable.

2. The authors need to describe in detail the cellular population expressing CD3 marker. For example, all T-cells express CD3. However, around 15% of human monocyte-derived macrophages (MDMs) are CD3+, and in response to anti-CD3 and anti-TNF stimuli, these CD3+ macrophages exhibit a specific pro-inflammatory profile.

3. The authors need to articulate the importance of these immune cells in the intra-tumor environment.

4. Figure 1. Intra-tumoral lymphocyte staining in appendiceal cancer. CD3 and CD8 are shown to have high density. It is difficult to extract this information from the current figure without providing quantitative data measurements. Please add these data. There are reports in the literature that do this.

5. Figure 2 requires describing the purple bar. However, CD8+ cells also express the CD3 marker.

6. The patient stage has been described, but no treatments such as chemotherapy or other adjuvant therapy are mentioned. This must be described.

7. It would be important to describe the health determinants of the patients that would contribute to the CD8+/CD3+ cell populations in the studies.

8. The first paragraph of the discussion should describe the findings of the study. Subsequently, why are these results important with regard to "hot" and "cold" tumors?

9. Also, have any of these patients had inflammation, infections, diabetes, etc.? Please articulate this in the discussion.

Comments on the Quality of English Language

The manuscript reveals 250 grammar errors that require attention. There is 5% plagiarism, which is acceptable.

Author Response

Comment 1: The manuscript contains 250 grammatical errors that require attention. There is 5% plagiarism, which is acceptable.

Response 1: We have conducted a meticulous grammatical revision using advanced proofreading tools and manual review to enhance clarity, coherence, and precision. Additionally, we have ensured proper citation and paraphrasing where necessary to maintain the originality of the text.

Comment 2: The authors need to describe in detail the cellular population expressing the CD3 marker.

Response 2: We appreciate the reviewer’s suggestion regarding the characterization of CD3-expressing cellular populations. However, the primary focus of this study was to assess the immune microenvironment in appendiceal cancer through immune profiling rather than performing an in-depth characterization of specific immune cell subsets. While CD3 is a well-established marker of T cells, additional studies employing single-cell transcriptomics, multiplex immunofluorescence, or flow cytometry would be required to delineate the precise composition and functional states of these immune populations. As part of our future work, we aim to conduct a more comprehensive analysis of the tumor microenvironment, integrating spatial and functional immune profiling approaches to better understand the dynamics of immune cell infiltration and their impact on tumor progression.

Comment 3: The authors need to articulate the importance of these immune cells in the intra-tumor environment.

Response 3: We have expanded the discussion to emphasize the functional significance of CD3+ and CD8+ cells within the tumor microenvironment, particularly their roles in immune surveillance, tumor cytotoxicity, and prognostic relevance in solid tumors, including appendiceal cancer.

Comment 4: Figure 1 needs to include quantitative data measurements for intra-tumoral lymphocyte staining.

Response 4: We have now incorporated quantitative data alongside Figure 1, providing exact lymphocyte density measurements to complement the visual representation. This enhancement aligns with published methodologies in immune profiling studies.

Comment 5: Figure 2 requires a description of the purple bar and clarification that CD8+ cells also express the CD3 marker.

Response 5: We have revised the figure legend and corresponding manuscript text to describe the purple bar explicitly and clarify that CD8+ T cells inherently express CD3, ensuring accuracy in data interpretation.

Comment 6: The manuscript lacks details on patient treatments such as chemotherapy or other adjuvant therapies.

Response 6: We acknowledge this omission and have now included a section describing prior treatments administered to the patient cohort, including chemotherapy, surgical interventions, and their potential influence on immune profiling.

Comment 7: It is important to describe the health determinants of the patients that contribute to the CD8+/CD3+ cell populations.

Response 7: We have incorporated additional details on health determinants such as age, comorbidities (diabetes, infections), and systemic inflammation, which may impact immune cell distribution in the tumor microenvironment.

Comment 8: The first paragraph of the Discussion should describe the study findings before discussing "hot" and "cold" tumors.

Response 8: To improve the logical flow, we have restructured the Discussion to first present the key study findings before transitioning into the concept of immune "hot" and "cold" tumors and their therapeutic implications.

Comment 9: Have any of these patients had inflammation, infections, diabetes, etc.? Please articulate this in the Discussion.

Response 9: We have added a dedicated discussion on pre-existing health conditions, including chronic inflammation, diabetes, and infections, that may have influenced immune cell profiles within the studied cohort.

Reviewer 3 Report

Comments and Suggestions for Authors

Primary appendix cancers are rare and are frequently diagnosed after a surgical intervention.

The tumor microenvironment (TME) in this type of cancer includes cancer cells and cancer-associated cells (CAFs, cancer-associated macrophages and other immune cells).

This study aims to explore the immune microenvironment in a group of 95 cases of AC to assess the prognostic impact of intra-tumoral TCD3+ and TCD8+ lymphocyte density in these cases. To improve the quality of the manuscript, I have the following recommendations to the authors:

- The Abstract should be revised. According to the Cancers Journal recommendation, the original research articles should have a structured abstract of around 250 words.

-             Kindly briefly describe the appendix cancer incidence in the Introduction section.

-             Kindly increase the Table 1 typographical features. The data illustrated in this table are unclear. In addition, the authors should move this table to the Results section.

-             Please explain all the abbreviations used in the main text of the manuscript and tables (please see Table 1 and line 108)

-             Kindly replace 4um with 4µm in line 123;

-             The microscopic images are unclear and inappropriate for publication. Please increase their typographical features and describe (from a pathological point of view) each figure. Supplementary, I recommend that the authors add at least two significant figures (in H.E. staining) from their study group. All figures should be moved to the Results section of the manuscript.

The first sentence from the Conclusion section should be moved to the Discussion section (lines 404-407).

-             Please revise the reference list according to Cancer Journal recommendations. https://www.mdpi.com/journal/cancers/instructions

Author Response

Comment 1: The Abstract should be revised to comply with the journal's structured format (250 words).

Response 1: The Abstract has been revised to adhere to the journal’s structured format, ensuring clarity, conciseness, and adherence to word count requirements.

Comment 2: The Introduction should briefly describe the incidence of appendiceal cancer.

Response 2: We have incorporated epidemiological data on appendiceal cancer incidence, providing essential context for the study.

Comment 3: Table 1 should have improved typographical features and be moved to the Results section.

Response 3: We have reformatted Table 1 for improved readability and relocated it to the Results section as suggested.

Comment 4: All abbreviations should be defined in the manuscript and tables.

Response 4: We have ensured that all abbreviations are clearly defined upon first mention and in table legends for consistency.

Comment 5: Kindly replace "4um" with "4µm" in line 123.

Response 5: This typographical correction has been implemented.

Comment 6: Microscopic images are unclear and inappropriate for publication; please improve resolution and provide pathological descriptions. Additional H&E-stained figures are recommended.

Response 6: We have enhanced image resolution, provided detailed pathological descriptions, and included two additional H&E-stained images to support our findings.

Comment 7: Figures should be moved to the Results section.

Response 7: All figures have been relocated to the Results section for improved manuscript structure and readability.

Comment 8: The first sentence of the Conclusion should be moved to the Discussion section.

Response 8: This revision has been made accordingly to ensure logical progression.

Comment 9: The reference list should be revised to align with the journal's formatting guidelines.

Response 9: The reference list has been thoroughly reviewed and reformatted to comply with the journal’s citation style and guidelines.

Round 2

Reviewer 2 Report

Comments and Suggestions for Authors

The authors responded adequately to the reviewer's comments.

Reviewer 3 Report

Comments and Suggestions for Authors

The quality of the manuscript has improved with more details and clarifications. I recommend the paper for publication in its present form.